# Characteristics of Hospitals Employing Dentists, and Utilization of Dental Care Services for Hospitalized Patients in Japan: A Nationwide Cross-Sectional Study

**DOI:** 10.3390/ijerph19116448

**Published:** 2022-05-26

**Authors:** Miho Ishimaru, Kento Taira, Takashi Zaitsu, Yuko Inoue, Shiho Kino, Hideto Takahashi, Nanako Tamiya

**Affiliations:** 1Department of Oral Health Promotion, Graduate School of Medical and Dental Sciences, Tokyo Medical and Dental University, Bunkyo 113-8549, Tokyo, Japan; zaitsu.ohp@tmd.ac.jp (T.Z.); inoue.ohp@tmd.ac.jp (Y.I.); shiho.kino.ohp@tmd.ac.jp (S.K.); 2Department of Health Services Research, Faculty of Medicine, University of Tsukuba, Tsukuba 305-8577, Ibaraki, Japan; ktaira.seiryokai@gmail.com (K.T.); ntamiya@md.tsukuba.ac.jp (N.T.); 3National Institute of Public Health, Wako 351-0197, Saitama, Japan; takahashi.h.aa@niph.go.jp

**Keywords:** dental care, hospitalized patients, multidisciplinary collaboration

## Abstract

Dental care for hospitalized patients can improve nutritional status and feeding function while reducing complications. However, such care in Japan is not uniformly provided. This investigation examined the presence and characteristics of hospitals where dentists work and the collaboration between medical and dental teams. This cross-sectional study involves 7205 hospitals using the administrative reports on the Hospital Bed Function of 2018. Indicators described were the proportion of hospitals employing dentists, those providing perioperative oral care, and those with a nutrition support team (NST) that included dentists. A two-level logistic regression model was performed using hospital-based and secondary medical area-based factors to identify factors associated with hospitals employing dentists and dental care services. Some hospitals had poor medical and dental collaboration, even those with dentists, and no-dentist hospitals had rare medical and dental collaboration. Factors positively associated with hospitals that employed dentists were diagnosis-procedure-combination-hospital types, the Japanese government-established hospitals compared with hospitals established by public organizations, among others. In conclusion, the present study found poor medical and dental collaboration was observed in some hospitals and that hospital type, region, and hospital founders were associated with the performance of collaborative medical and dental care.

## 1. Introduction

Dental care contributes to improving the general condition of hospitalized patients and preventing adverse events. Previous studies have reported that preoperative oral care reduces postoperative and ventilator-associated pneumonia and mortality [1,2,3,4,5]. A recent study found that oral care, swelling rehabilitation, and other types of oral management by dental hygienists resulted in improved nutritional status and shortened recuperation in the hospital [6].

Not all centers in Japan provide sufficient dental care services to hospitalized patients. A previous report from the Japan Dental Association documented that only 26.1% of hospitals had a department of dentistry or oral surgery [7]. Approximately half of the hospitals without such departments did not provide any routine oral care by medical staff. Furthermore, patients did not receive adequate oral care even in hospitals with dentists. According to a questionnaire study in academic hospitals, 72.2% of patients had oral problems at discharge, but only 18.2% of them were seen by a dentist during their hospitalization [8].

Hospital quality indicators such as the number of specialists providing any care or treatment are essential to improving the quality of in-patient care [9,10]. However, to the best of our knowledge, few studies have delineated the indicator focusing on the availability of dentists in hospitals and the multidisciplinary collaboration for perioperative oral care and nutrition support for in-hospital patients.

This study aimed to describe the presence and activities of dental services in hospitals in Japan and to explore hospital-related organizational factors that affect dental care and the collaboration between medical and dental teams.

## 2. Materials and Methods

### 2.1. Data Sources

This was a cross-sectional study conducted by secondary use of the hospital function administrative reports of 2018. The survey data were available to the public on the Japanese Ministry of Health, Labor, and Welfare website [11]. The reporting system was initiated in 2014 to provide data for facilitating the vision of regional medical care [12]. All hospitals in Japan are obligated to report on their function, including their clinical departments, the number of medical staff and dentists, the number of surgical operations performed, and the number of specific procedures performed annually. No blinding of the hospital name or address was carried out in the survey data.

### 2.2. Inclusion and Exclusion Criteria

The data from 2018 of all reported hospitals were included in this study. Exclusion criteria were hospitals that established or closed in 2017, hospitals that did not report the number of dentists working at their institutions, and dental hospitals. Dental hospitals were defined as hospitals where working dentists were greater than the number of working physicians. The number of hospitals meeting the inclusion criteria was 7353, and 167 hospitals were excluded based on the exclusion criteria (Figure 1).

### 2.3. Variables

Hospitals employing dentists were defined as those reporting the number as >0.1 dentists/week. From the Hospital Bed Function data, we extracted information regarding collaboration across medical and dental services through the activity of the nutrition support team (NST), which variously involved dentists and perioperative oral care for surgical patients. The NST intervention is implemented to improve the nutritional status of those with malnutrition by performing dental check-ups and treatments such as scaling, treating dental caries, and making dentures if needed. When dentists join the NST, the additional cost can be reimbursed for hospitalized patients. Perioperative oral care (dental check-ups and treatment) can be reimbursed for patients undergoing certain surgical procedures under general anesthesia. Dentists involved in medical–dental collaborations are either hospital-employed dentists or dentists who practice at a community dental clinic.

The hospital factors included surgery hospitals, the type of diagnosis procedure combination (DPC), the hospital’s founder, functions, and the number of working dentists. 

We defined surgical hospitals as hospitals that perform any type of surgery under general anesthesia. The DPC is a case-mix patient classification system developed in Japan in 2002. This system is linked with a lump-sum payment for hospitalized patients in acute care hospitals, called the “DPC per-diem payment system” [13,14]. According to the criteria for DPC-eligible hospitals by the Ministry of Health, Labor, and Welfare, hospitals are accredited in one of the following four groups: I, II, III, and non-DPC. Group I DPC hospitals provide the highest level of medical care and include academic hospitals. Group II DPC hospitals have functions that correspond to those of university hospitals. Group III DPC hospitals provide standard acute care [14]. The hospital establishers were stratified into the Japanese government, universities, public organizations, social insurance-related organizations, medical corporations, other corporations, and others (companies, individuals, and consumers’ co-operatives). Hospital functions included regional medical care support, emergency care, recovery rehabilitation, and long-term care. The regional factors were the presence of a dental college, a faculty of dentistry, and the number of dental clinics per 10,000 population in each secondary medical area. The secondary medical areas were utilized as the geographic unit because they were the administrative areas for the community medical plan [15]. In Japan, there are 344 secondary medical areas. The regional divisions were categorized into seven areas: Hokkaido, Tohoku, Kanto, Chubu, Kinki, Chugoku/Shikoku, and Kyushu/Okinawa. 

### 2.4. Statistical Analysis

The following three indicators were examined: hospitals employing dentists, hospitals with an NST including dentists, and hospitals providing perioperative oral care for patients undergoing surgical operations. We described the number of hospitals employing dentists for each hospital accreditation and geographic factor. Stratified by the hospital employing or not-employing dentists, hospitals with an NST including dentists were described. Limited to surgical hospitals, we described the hospitals providing perioperative oral care for patients undergoing surgical operations stratified by the hospital employing or not-employing dentists. To quantify the association between hospital factors, regional factors, and the three indicators defined above, we performed a two-level logistic regression with a random intercept. The multilevel models included the binomial outcome variable (hospitals employing dentists, hospitals with an NST including dentists, or hospitals providing perioperative oral care), cluster variable (secondary medical area), hospital factors, and regional factors. The hospital factors were the type of DPC, the hospitals’ establishers, and functions. The regional factors were regional divisions, the number of dental clinics per 10,000 population, and the presence of a dental college or a faculty of dentistry. The odds ratio and the 95% confidence interval were calculated. All statistical analyses were conducted using R (version 3.6.1; R Foundation for Statistical Computing, Vienna, Austria). 

## 3. Results

A total of 7205 hospitals were included in this study. Those employing dentists were 1584 (22.0%), of which 324 (20.5%) have an NST including dentists. Among the hospitals not employing dentists, 50 (0.9%) have an NST including dentists. Out of the 3279 performing operations with general anesthesia, 689 (21.0%) provided perioperative oral care for patients. This included 598/1076 (55.6%) of the surgery hospitals employing dentists and 91/2203 (4.1%) of those not employing dentists. Regarding the DPC type, the highest prevalence of hospital-working dentists was in group I (100.0%) and decreased, respectively, through groups II to III, reaching a low of 14.3% in the non-DPC hospitals. Further hospital characteristics are shown in Table 1 and Table 2.

Figure 2 shows the odds ratios of factors associated with hospitals that employ a dentist. DPC-hospital types were strongly associated with hospitals employing dentists compared to non-DPC hospitals. In terms of the hospital establishers, medical corporations were the least likely to be associated with hospitals employing dentists compared to hospitals established by the government. Regional medical care support was the function most likely to be associated with the hospitals employing dentists. In terms of regional divisions, Tohoku, Kanto, and Chubu were significantly more likely to be associated with hospitals employing dentists than Hokkaido.

The associations between hospitals providing NST support, including dentists and each hospital characteristic, as stratified by the number of hospitals employing/not employing dentists, are shown in Figure 3. The DPC-hospital types were strongly associated with providing an NST that included dentists in both hospitals employing and not employing dentists. Furthermore, regional medical support hospitals were more likely than others to provide an NST that included dentists while long-term care hospitals were less likely to be associated with this for hospitals employing dentists.

The associations between hospitals performing perioperative oral care and each hospital characteristic, as stratified by whether the hospitals employed dentists, are shown in Figure 4. The DPC-hospital types were strongly associated with performing perioperative oral care in both hospitals employing and not employing dentists. Moreover, hospitals employing dentists, being established by a social insurance-related organization, functioning as a regional medical care support or emergency hospital, and having dental colleges or a faculty of dentistry were positively associated with performing perioperative oral care. In contrast, the regions Tohoku and Kyusyu/Okinawa were negatively associated with performing perioperative oral care when compared to Hokkaido for hospitals employing dentists.

## 4. Discussion

This study examined the presence and characteristics of hospitals that influence the adequate provision of in-hospital dental care, including collaborative efforts across medical and dental services. The proportion of total hospitals employing dentists was 22.0%, while the proportion of hospitals with an NST that included dentists was 4.5%. Perioperative oral care was performed in 21.5% of surgical hospitals. The DPC-hospital types were strongly associated with employing dentists and promoting joint dental and medical care efforts.

According to a report from the Japan Dental Association, 26.1% of hospitals had a department of dentistry or oral surgery, 22.6% of hospitals employing dentists provided an NST, and 33.1% of hospitals employing dentists provided perioperative oral care [7]. In contrast, these percentages in our study were 22.0%, 20.5%, and 55.6%, respectively. Moreover, that report found that in hospitals not employing dentists, 6.0% provided an NST that included dentists and 20.8% provided perioperative oral care [7]. These percentages in our study were 0.9% and 4.1%, respectively. Because the report by the Japan Dental Association was a questionnaire study, the results might have been overestimated. Furthermore, the proportion of hospitals that perform perioperative oral care was higher than that of the previous study [7]. This discrepancy may be due to the previous study including all hospitals to establish this proportion, while the present study utilized only those hospitals performing the eligible surgery. The method used in our study can obtain a more accurate percentage.

In terms of the hospital characteristics, the DPC-hospital type was strongly associated with hospitals employing dentists and providing collaboration across medical and dental care. The DPC type is a Japan-specific classification that comprehensively represents the function of an acute hospital. This includes the hospital system, the number of nurses, the number of junior residents, and any incentives for the roles of medical institutions [13]. This study found that cooperative medical and dental service interactions occurred more often in more advanced hospitals.

The hospital founders can be divided into public (the Japanese government, public organizations, social insurance-related organizations), private (medical corporations, other corporations, others), and mixed (university) sources. Our study, therefore, implied that public hospitals might be more likely to have working dentists and provide perioperative oral care than private hospitals. This may suggest that private hospitals are less interested in medical–dental collaboration because it is not currently profitable. The implementation of an NST that includes dentists costs only 500 Japanese yen (4.5 US dollars) per patient, while the cost of perioperative oral care is about 10,720 Japanese yen (97.5 US dollars) per hospitalized patient [16]. This shows that the cost of dental care may not be high enough to be performed regularly. Appropriate costs for medical and dental collaboration that are profitable for hospitals may be important in increasing the number of hospitals engaging in efforts to bring together their medical and dental services.

Of those institutions that employ dentists, regional medical care support hospitals were significantly more likely to be associated with having a dentist-invested NST and providing perioperative oral care. One role of regional medical care support hospitals is to provide care to patients on referral, including the reverse referral in which patients are referred to family doctors. Therefore, these hospitals may have a system that facilitates referrals to dentists and physicians. Concerning regional factors, having dental colleges or a faculty of dentistry was associated with the performance of perioperative oral care in hospitals employing dentists. Reasons for this may include education on the importance of providing oral care and the newly graduated dental residents being more inclined to work at nearby hospitals.

The main finding that our study highlighted is that some hospitals had poor medical and dental collaboration, even those with dentists. Furthermore, most hospitals not employing dentists did not provide any NST support that included dentists or perioperative oral care for hospitalized patients. The current Japanese policy encourages medical–dental collaborations in hospitals; however, this has not yet been sufficiently achieved. In the hospitals that employ dentists, two explanations for poor collaboration can be provided. Firstly, dentists working in hospitals may have been maxillofacial and oral surgeons, thus limiting their time to provide dental care services. Secondly, a system may not be available through which a physician can easily request a dentist in the same hospital to perform perioperative oral care or join an NST. For hospitals not employing dentists, there may be a need for the creation a system that works closely with community dental clinics because the dentists with community dental clinics only provide the dental care for in-hospital patients there. Although there are sufficient community dental clinics in Japan [17,18], they are rarely involved in the care of hospitalized patients. More dental professionals should be involved in the care of these patients, which will lead to improvements in the oral health of hospitalized patients overall. Further studies are needed to develop more specific programs to reduce the inequality in hospitalized patients’ access to dental care for hospitals.

To the best of our knowledge, the present study was the first to describe the presence and characteristics of hospitals regarding whether they employ dentists, have an NST that includes dentists, or provide perioperative oral care. One strength of our research was that almost all hospitals in Japan were required to answer the administrative reports used as our database. We, therefore, believe that our study has little selection bias. Thus, we were able to elucidate from our research that the availability of collaborative medical and dental care to patients may depend on the admitting hospital. Furthermore, according to the OECD Health Data 2020, Japan’s average length of stay in an acute hospital (16.0 days) is outstandingly long compared to other OECD countries (7.5 days in Germany, 6.2 days in the UK, 5.5 days in the US) [19]. The average stay in rehabilitation hospitals, to which some patients are admitted after discharge from an acute hospital, is 70.0 days [20]. Therefore, it is not enough to treat dental problems only after the patient is discharged from the hospital; it is very important that the hospital provides an environment in which professional oral care can be provided during hospitalization and when patients are in poor general condition after major surgery.

According to Donabedian’s health care quality model, improvement in care structure should improve clinical processes and patient outcomes [21]. We suggest whether dental care is provided for in-hospital patients may be one of the new hospital quality indicators. A previous study reported that dental and oral care is a quality indicator for the aged population [22]. Patients cannot choose to be admitted to hospitals with dental care services if they need dental care because most of them are admitted to hospitals to treat severe general conditions. It is important to reduce the inequality in hospitals to provide dental care, which may lead to inequality in hospitalized patients’ access to this more comprehensive care if ignored. Appropriate dental interventions would contribute to preventing the onset of pneumonia [1,2,3,4,5], improving the nutritional status [6], and maintaining the oral condition of hospitalized patients. Few reports describe dental care for in-hospital patients in any country. Therefore, we believe that describing the presence and activity of dental utilization for in-hospital patients in Japan nationwide is important to help readers understand the current situation.

This study has some limitations. Firstly, the administrative reports used in this study were based on hospital surveys. Although all hospitals were required to respond, some hospitals did not. Moreover, data were only available for dental practices for those performed in June 2017. The hospitals may have been misclassified if no patient underwent dental procedures that month.

## 5. Conclusions

We found that some hospitals with dentists also lack medical–dental coordination, and many hospitals do not employ dentists; they do not provide NST support, including dentists or perioperative oral care, for hospitalized patients. Furthermore, the DPC-hospital type, hospital establisher, hospital function, and regional division were associated with whether hospitals employed dentists and performed collaborative medical and dental care. Further study is needed to develop more specific programs to reduce the inequality in hospitalized patients’ access to dental care in hospitals.

## Figures and Tables

**Figure 1 ijerph-19-06448-f001:**
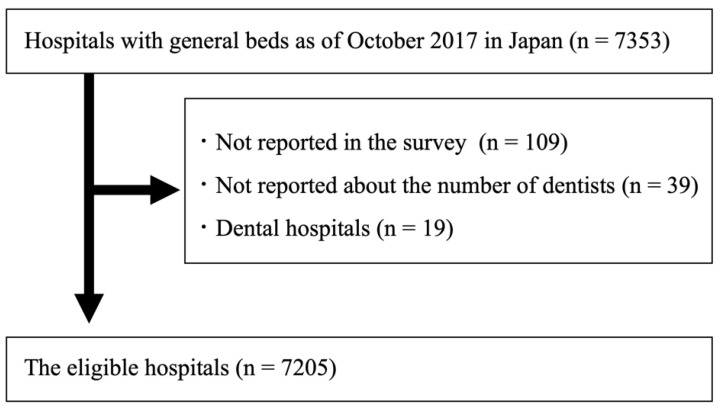
Flow chart of the eligible hospitals in the present study.

**Figure 2 ijerph-19-06448-f002:**
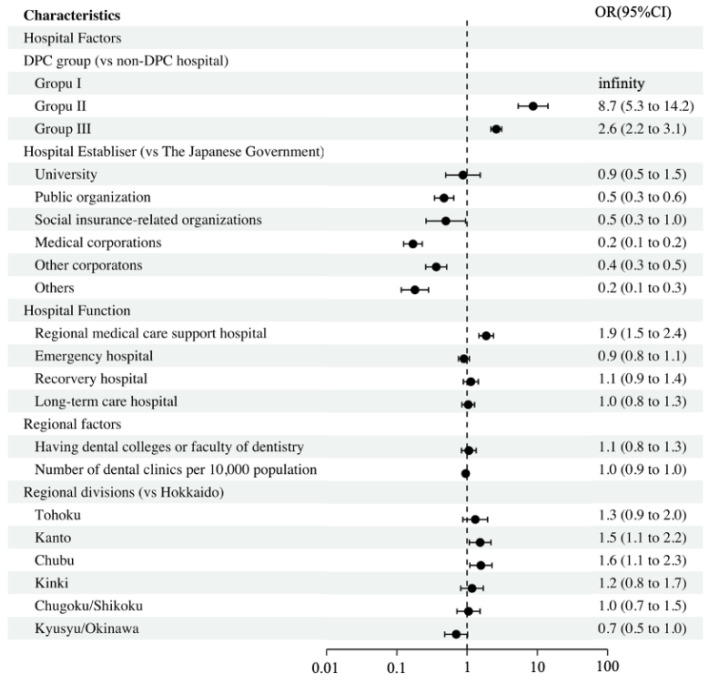
Factors associated with hospitals employing dentists. Abbreviations: OR, odds ratio; 95% CI, 95% confidence interval.

**Figure 3 ijerph-19-06448-f003:**
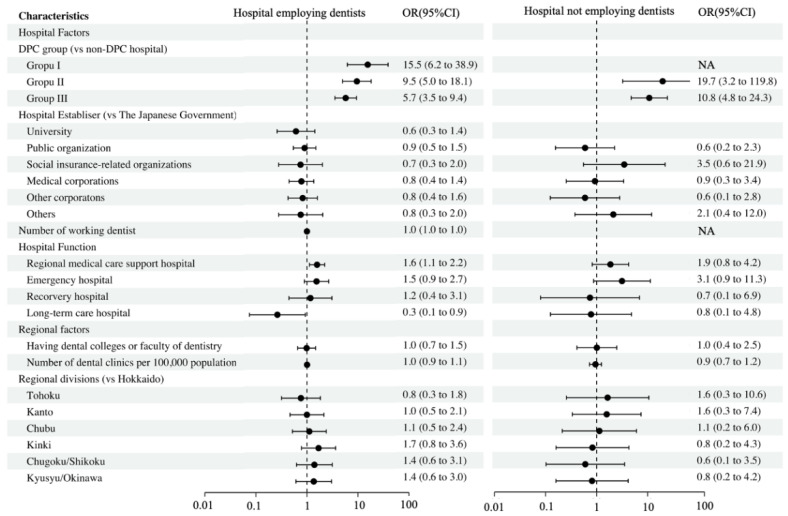
Factors associated with hospitals providing a nutrition support team based on dentist employment status. Abbreviations: OR, odds ratio; 95% CI, 95% confidence interval.

**Figure 4 ijerph-19-06448-f004:**
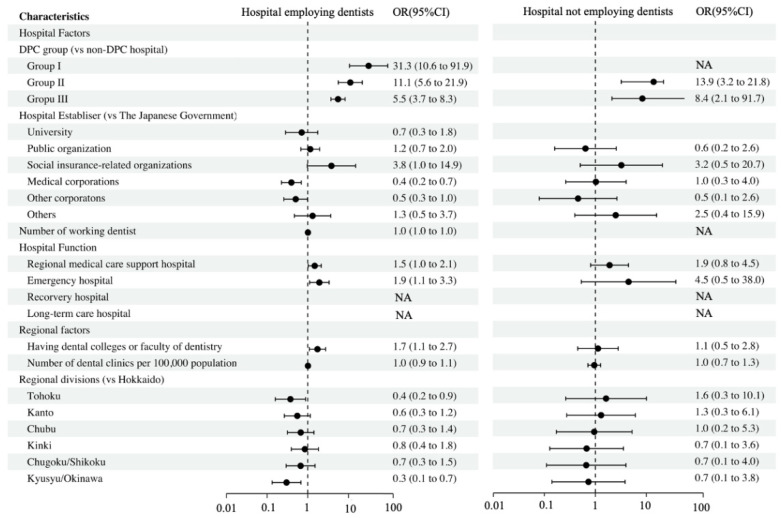
Factors associated with hospitals providing perioperative oral care based on dentist employment status. Abbreviations: OR, odds ratio; 95% CI, 95% confidence interval.

**Table 1 ijerph-19-06448-t001:** Characteristics of hospitals with/without dentists and having a nutrition support team including dentists.

		Hospital with Dentists				Hospital without Dentists			
	Total			Nutrition Support Team Including Dentists				Nutrition Support Team Including Dentists	
	N	n	(%)	n	(%)	N	(%)	n	(%)
N	7205	1584	(22.0)	324	(20.5)	5621	(78.0)	50	(0.9)
**Hospital factors**									
**DPC type (%)**									
Group I	82	82	(100.0)	37	(45.1)	0	(0.0)	-	-
Group II	140	113	(80.7)	55	(48.7)	27	(19.3)	2	(7.4)
Group III	1442	595	(41.3)	198	(33.3)	847	(58.7)	37	(4.4)
Non-DPC	5541	794	(14.3)	34	(4.3)	4747	(85.7)	11	(0.2)
**Establisher of hospitals (%)**									
The Japanese government	226	133	(58.8)	32	(24.1)	93	(41.2)	3	(3.2)
University	162	127	(78.4)	43	(33.9)	35	(21.6)	0	0.0
Public organizations	1166	486	(41.7)	146	(30.0)	680	(58.3)	10	(1.5)
Social insurance-related organizations	53	26	(49.1)	8	(30.8)	27	(50.9)	3	(11.1)
Medical corporations	4722	605	(12.8)	63	(10.4)	4117	(87.2)	27	(0.7)
Other corporations	577	165	(28.6)	25	(15.2)	412	(71.4)	4	(1.0)
Others	299	42	(14.0)	7	(16.7)	257	(86.0)	3	(1.2)
**Regional medical care support hospital**	554	342	(61.7)	147	(43.0)	212	(38.3)	13	(6.1)
**Emergency hospital**	3770	1073	(28.5)	299	(27.9)	2697	(71.5)	46	(1.7)
**Recovery rehabilitation hospital**	735	114	(15.5)	6	(5.3)	621	(84.5)	1	(0.2)
**Long-term care hospital**	1729	238	(13.8)	3	(1.3)	1491	(86.2)	2	(0.1)
**Number of dentists working in a hospital (mean (SD))**	1.4 (13.5)	6.3	(28.2)	7.0	(19.6)	-	-	-	-
**Regional factors**									
Having dental colleges or faculty of dentistry (%)	1437	310	(21.6)	66	(21.3)	1127	(78.4)	11	(1.0)
Number of dental clinics per 10,000 population (mean (SD))	53.5(15.4)	53.9	(20.9)	55.6	(24.5)	53.5	(15.36)	53.96	(11.61)
Regional divisions (%)									
Hokkaido	489	88	(18.0)	11	(12.5)	401	(82.0)	3	(0.7)
Tohoku	476	135	(28.4)	19	(14.1)	341	(71.6)	4	(1.2)
Kanto	1758	438	(24.9)	79	(18.0)	1320	(75.1)	17	(1.3)
Chubu	1007	292	(29.0)	65	(22.3)	715	(71.0)	6	(0.8)
Kinki	1217	256	(21.0)	68	(26.6)	961	(79.0)	8	(0.8)
Chugoku/Shikoku	955	186	(19.5)	41	(22.0)	769	(80.5)	4	(0.5)
Kyushu/Okinawa	1303	189	(14.5)	41	(21.7)	1114	(85.5)	8	(0.7)

**Table 2 ijerph-19-06448-t002:** Characteristics of surgery hospitals with/without dentists and providing perioperative oral care.

	Surgery Hospitals	With Dentists			Without Dentists		
			Perioperative Oral Care			Perioperative Oral Care	
	N	N	n	(%)	N	n	(%)
N	3279	1076	598	(55.6)	2203	91	(4.1)
**Hospital factors**							
**DPC type (%)**							
Group I	82	82	74	(90.2)	-	-	-
Group II	139	112	94	(83.9)	27	8	(29.6)
Group III	1373	583	377	(64.7)	790	70	(8.9)
Non-DPC	1685	299	53	(17.7)	1386	13	(0.9)
**Establisher of hospitals (%)**							
The Japanese government	171	101	57	(56.4)	70	7	(10.0)
University	148	117	94	(80.3)	31	1	(3.2)
Public organizations	827	423	285	(67.4)	404	40	(9.9)
Social insurance-related organizations	48	24	21	(87.5)	24	6	(25.0)
Medical corporations	1681	295	84	(28.5)	1386	26	(1.9)
Other corporations	292	90	39	(43.3)	202	7	(3.5)
Others	112	26	18	(69.2)	86	4	(4.7)
**Regional medical care support hospital**	543	338	254	(75.1)	205	44	(21.5)
**Emergency hospital**	2713	948	560	(59.1)	1765	86	(4.9)
**Recovery rehabilitation hospital**	43	8	0	0.0	35	0	0.0
**Long-term care hospital**	33	13	0	0.0	20	0	0.0
**Number of dentists working in a hospital (mean (SD))**	2.3(15.5)	7.0(26.5)	9.1	(29.2)	-	-	-
**Regional factors**							
Having dental colleges or faculty of dentistry (%)	665	209	139	(66.5)	456	21	(4.6)
Number of dental clinics per 10,000 population (mean (SD))	54.6(19.8)	52.4 (17.4)	56.7	(27.0)	54.49 (18.0)	53.55	(12.3)
Regional divisions (%)							
Hokkaido	200	51	28	(54.9)	149	2	(1.3)
Tohoku	210	86	41	(47.7)	124	8	(6.5)
Kanto	902	306	162	(52.9)	596	15	(2.5)
Chubu	463	217	135	(62.2)	246	11	(4.5)
Kinki	626	183	114	(62.3)	443	25	(5.6)
Chugoku/Shikoku	374	116	67	(57.8)	258	14	(5.4)
Kyushu/Okinawa	504	117	51	(43.6)	387	16	(4.1)

Note: The odds ratio of DPC group I hospitals rather than non-DPC hospitals was infinity because all DPC group I hospitals employed dentists.

## Data Availability

The datasets generated and/or analyzed during the current study are available on the Japanese Ministry of Health, Labor, and Welfare website, (https://www.mhlw.go.jp/stf/seisakunitsuite/bunya/open_data_00002.html, accessed on 20 May 2020).

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
