# Peer review of "Characteristics of Hospitals Employing Dentists, and Utilization of Dental Care Services for Hospitalized Patients in Japan: A Nationwide Cross-Sectional Study"

_ijerph, 2022, doi:10.3390/ijerph19116448_

Round 1

Reviewer 1 Report

Methodology and results should be simplified and further clarified for the readers.

Extensive language editing needed. As a result, certain areas of the text are incomprehensible. (i.e. In terms of the factors associated with hospitals that employ dentists and provide collaboration across medical and dental care, the present study revealed the DPC-hospital type was the feature strongly associated with these parameters.)

Author Response

Response: Thank you for your comments. We have revised the methodology and results section for clarity and readability.(for example, lines 129-134, and 203-205)

Reviewer 2 Report

The article presents a topic of interest, since dental coverage for hospitalized patients is a controversial issue at the international level. Below I present certain doubts and suggestions that I believe will help to better organize and understand the article.

You specify that the objective is to identify and analyze the characteristics of hospitals where dentists work, including the collaboration between the collaboration between medical and dental teams. I suggest changing the objective to evaluate the presence and activity of dental services in hospitals in Japan, since not only hospitals with dental coverage were analyzed. In the material and methods section, it is not specified how the data collection was carried out (number of operators, coding of hospitals, blinding of the study), which is very important from the methodological point of view.

In addition, it would be of great interest to detail dental activities in hospitals, since it is not clear if it is exclusively preventive or if dental treatments are also included. I am also wondering which medical team performed the perioperative oral treatment in those hospitals without dentists.

The statistics section must include specification of which statistical methods were carried out and with what level of confidence.

It is recommended to increase the number of bibliographic references used in order to increase the scientific basis and more solidly support the results of the study.

Author Response

The article presents a topic of interest, since dental coverage for hospitalized patients is a controversial issue at the international level. Below I present certain doubts and suggestions that I believe will help to better organize and understand the article. 

  1. You specify that the objective is to identify and analyze the characteristics of hospitals where dentists work, including the collaboration between the collaboration between medical and dental teams. I suggest changing the objective to evaluate the presence and activity of dental services in hospitals in Japan, since not only hospitals with dental coverage were analyzed.

Response: Thank you for your comments. According to your suggestion, we have revised the objective of the study to evaluate the presence and activity of dental services in hospitals in Japan in the Introduction and the Discussion sections. (lines 52 and 184)

 In the material and methods section, it is not specified how the data collection was carried out (number of operators, coding of hospitals, blinding of the study), which is very important from the methodological point of view. 

Response: Our study was conducted using secondary data from the hospital bed function survey. All hospitals in Japan are obligated to report on their function. Anybody can access the survey data publicly on the website. No blinding of the hospital name or address was carried out in the survey data. We have added the explanation about the data collection in the methods section. (Lines 58 and 64)

In addition, it would be of great interest to detail dental activities in hospitals, since it is not clear if it is exclusively preventive or if dental treatments are also included.

Response: The perioperative oral care and nutrition support team activity generally included preventive dental care and treatment such as scaling, treating dental caries, and making dentures if patients needed any dental treatment. (Lines 83 and 84)

I am also wondering which medical team performed the perioperative oral treatment in those hospitals without dentists. 

Response: In hospitals without dentists, preoperative oral care was performed by dentists working in community dental clinics close to the hospitals. (Lines 240-243)

The statistics section must include specification of which statistical methods were carried out and with what level of confidence. 

Response: We conducted two-level logistic regression analyses and calculated the odds ratio and the 95% confidence interval. We have revised the statistics section in response to your comments. (Line 125)

It is recommended to increase the number of bibliographic references used in order to increase the scientific basis and more solidly support the results of the study.

Response: We have included four references (9,10,19, and 20) in the revised manuscript to support the importance of the study.

Reviewer 3 Report

Isn't the same study being performed in other areas worldwide?
If this is the case, no explanation was provided in the introduction or discussion. If not, it appears that it is preferable to be published in Japanese journals.

Author Response

Response: Thank you for your comments. We have added texts about hospital quality indicators such as providing specific treatment or care for inpatients and included the four references (9,10,19, and 20). The national medical system of hospitals employing dentists or providing dental care for in-hospital patients may differ among countries. We believe that describing the presence and activity of dental utilization for in-hospital patients in Japan nationwide is important to help readers understand the current situation in other countries. We have explained these in the sentences added to the Discussion section. (Lines 267-270)

Reviewer 4 Report

Ishimaru et. al. aim to reduce the inequality in hospitalized patients' access to dental care. They explored the data from 7205 hospitals (of the year 2018) in the manuscript and found some variables, such as hospital type, established, function, and location, were associated with the service that the hospitals provided.

The authors should provide more background information to justify the research. For example, why would exploring the characteristics of hospitals with or w/o dental care would help the situation? From my understanding, if a patient needs dental care, it is more reasonable for the patient to look for a hospital with dental care services, instead of figuring out why a particular hospital does not provide such services. Or, what is the causality reason that a hospital does not provide dental care services? Is it because there is already a hospital with dental care services nearby? Without this background information, I cannot justify the merit of the manuscript.

Author Response

Ishimaru et. al. aim to reduce the inequality in hospitalized patients' access to dental care. They explored the data from 7205 hospitals (of the year 2018) in the manuscript and found some variables, such as hospital type, established, function, and location, were associated with the service that the hospitals provided.

The authors should provide more background information to justify the research. For example, why would exploring the characteristics of hospitals with or w/o dental care would help the situation? From my understanding, if a patient needs dental care, it is more reasonable for the patient to look for a hospital with dental care services, instead of figuring out why a particular hospital does not provide such services.

Response: Thank you for your comments. As pointed out, outpatients can visit dental clinics or the department of general dentistry in the hospitals if they need dental care. However, in-hospital patients cannot select admission hospitals with dental care services if they need dental care because most of them admit choosing facilities to treat their severe general conditions, such as cancer, stroke, cardiovascular disease, fracture, or other diseases. They cannot select the hospitals focusing on them with or without dental care utilization. The purpose of our study was not to help patients themselves choose an admission hospital with dental care but to clarify that in-hospital patients cannot receive adequate dental care because some of these hospitals do not provide dental care. We have added some explanations to the Discussion section (lines 256-262)

Or, what is the causality reason that a hospital does not provide dental care services? Is it because there is already a hospital with dental care services nearby? Without this background information, I cannot justify the merit of the manuscript.

Response: The reason a hospital does not provide dental care services remains unknown. Therefore, our study explored the factor related to hospitals employing dentists and providing collaboration between medical and dental care services. Our study found that hospital function and hospital founders were related to hospitals employing dentists. Furthermore, hospitals located in areas having dental colleges or a faculty of dentistry were positively associated with performing perioperative oral care. Therefore, the reason a hospital does not provide dental care services may be considered hospital function, cost, or location that makes it easy to connect the dental professionals. This finding helps explore methods to encourage hospitals that do not provide dental care to provide dental care in further research. We have added a sentence in the Discussion section. (lines 246-248)

Round 2

Reviewer 2 Report

The research is interesting. Although it has methodological limitations, these cannot be corrected due to the nature of the study. The study clearly shows the problem of the absence of dentists and/or dental care for hospital patients.  

Author Response

The research is interesting. Although it has methodological limitations, these cannot be corrected due to the nature of the study. The study clearly shows the problem of the absence of dentists and/or dental care for hospital patients.  

Response: We wish to express our appreciation to the reviewers for their insightful comments on our paper. As you indicated your comments, the present study has some limitations. However, this is the first study to evaluate the problem of the absence of dentists and/or dental care for hospital patients. We believe that our findings offer essential information to help hospitalized patients receive dental care in the future.

Reviewer 4 Report

Thanks for responding to my previous comments. However, those responses do not clear my questions. Specifically,

1. The authors mentioned in 

"

However, in-hospital patients cannot select admission hospitals with dental care services if they need dental care because most of them admit choosing facilities to treat their severe general conditions, such as cancer, stroke, cardiovascular disease, fracture, or other diseases. They cannot select the hospitals focusing on them with or without dental care utilization. The purpose of our study was not to help patients themselves choose an admission hospital with dental care but to clarify that in-hospital patients cannot receive adequate dental care because some of these hospitals do not provide dental care.

"

that the manuscript was to clarify that in-hospital patients cannot receive adequate dental care because some hospitals do not provide such services. In my opinion, dental issues are rarely life-threatening causes to patients. Those already admitted patients can be transferred to a dental office if the life-threatening event has gone. I don't get the point why the lack of dental services needs to be clarified.

2. The authors mentioned in

``

The reason a hospital does not provide dental care services remains unknown. Therefore, our study explored the factor related to hospitals employing dentists and providing collaboration between medical and dental care services. Our study found that hospital function and hospital founders were related to hospitals employing dentists. Furthermore, hospitals located in areas having dental colleges or a faculty of dentistry were positively associated with performing perioperative oral care. Therefore, the reason a hospital does not provide dental care services may be considered hospital function, cost, or location that makes it easy to connect the dental professionals.

''

that the causality is unknown, and hence factors are investigated for correlations. While correlation may suggest some relationship among the data, I think it is inappropriate and dangerous to use such an approach in this study. In my opinion, it is critical to study the underlying policy cause of the assignment of dental services if one wishes to survey such information. Such policy causes may include, but not limited to, Are the hospitals public or private? If private, are they for-profit or not-for-profit? Are they regular hospitals or research hospitals (with grant support)? I think these questions are more important than the variables provided in the manuscript. Alternatively, there is a possibility that the authors are representing the authorities to investigate how to design new policies to encourage the hospitals to add dental services. If so, the authors should confirm in the manuscript that there are no government policy barriers that stop the hospitals from adding dental services.

Besides, the authors mentioned that " This finding helps explore methods to encourage hospitals that do not provide dental care to provide dental care in further research. ".

For now, I don't think the findings are helpful in that way. Maybe the authors can provide more background information. If a hospital is not allowed to provide dental services by certain authority/policy, or if a hospital is for-profit but adding dental services is not profitable because there are more affordable services nearby, or if a hospital is designed for certain specialty diseases that adding dental services is inherently not an interest of them, all such situations are quite likely where the manuscript's results won't help. Alternatively, the authors may filter their data to include only the hospitals without such mentioned concerns.

For the current version, I feel a significant amount of background information is still missing. The authors should work further to clarify the background information and to make sure the data and results reasonably support the conclusions.

Author Response

  1. The authors mentioned in "However, in-hospital patients cannot select admission hospitals with dental care services if they need dental care because most of them admit choosing facilities to treat their severe general conditions, such as cancer, stroke, cardiovascular disease, fracture, or other diseases. They cannot select the hospitals focusing on them with or without dental care utilization. The purpose of our study was not to help patients themselves choose an admission hospital with dental care but to clarify that in-hospital patients cannot receive adequate dental care because some of these hospitals do not provide dental care.”

that the manuscript was to clarify that in-hospital patients cannot receive adequate dental care because some hospitals do not provide such services. In my opinion, dental issues are rarely life-threatening causes to patients. Those already admitted patients can be transferred to a dental office if the life-threatening event has gone. I don't get the point why the lack of dental services needs to be clarified.

Response: Thank you for your insightful comments on our paper. To answer your question, I describe the characteristics of the Japanese hospitalization system below. According to the OECD Health Data 2020, Japan’s average length of stay in an acute hospital (16.0 days) is outstandingly long compared to other OECD countries. (7.5 days in Germany, 6.2 days in the UK, 5.5days in US) (https://data.oecd.org/healthcare/length-of-hospital-stay.htm). Furthermore, the average stay in rehabilitation hospitals to which some patients are admitted after discharge from an acute hospital is 70.0 days ( https://www.mhlw.go.jp/content/12404000/000823126.pdf, Japanese). Therefore, if in-hospital patients did not have dental care available, several months or years may pass by the time they return home and seek dental care after their life-threatening event.

Further, patients’ immune systems are compromised when they experience major surgery, and dental problems can significantly impact their overall condition. A previous study reported that professional oral care reduced the incidence of postoperative pneumonia and in-hospital mortality for cancer (reference 1) or cardiovascular patients (reference 2) and ventilator-associated pneumonia for ICU patients (reference3-5). These events are life-threatening. Another study revealed that intensive dental care by a dental hygienist for rehabilitation hospitalized patients improves their Activity in Daily Living and nutrition status (reference 6). Therefore, it is not enough to treat dental problems only after the patient is discharged from the hospital; it is very important that the hospital provides an environment in which professional oral care can be provided during hospitalization and when patients are in poor general condition. This content is presented in the Introduction and Discussion sections. We have added a further content to the Discussion section (lines 267-269).

  1. The authors mentioned in

“The reason a hospital does not provide dental care services remains unknown. Therefore, our study explored the factor related to hospitals employing dentists and providing collaboration between medical and dental care services. Our study found that hospital function and hospital founders were related to hospitals employing dentists. Furthermore, hospitals located in areas having dental colleges or a faculty of dentistry were positively associated with performing perioperative oral care. Therefore, the reason a hospital does not provide dental care services may be considered hospital function, cost, or location that makes it easy to connect the dental professionals.''

that the causality is unknown, and hence factors are investigated for correlations. While correlation may suggest some relationship among the data, I think it is inappropriate and dangerous to use such an approach in this study. In my opinion, it is critical to study the underlying policy cause of the assignment of dental services if one wishes to survey such information. Such policy causes may include, but not limited to, Are the hospitals public or private? If private, are they for-profit or not-for-profit? Are they regular hospitals or research hospitals (with grant support)? I think these questions are more important than the variables provided in the manuscript. Alternatively, there is a possibility that the authors are representing the authorities to investigate how to design new policies to encourage the hospitals to add dental services. If so, the authors should confirm in the manuscript that there are no government policy barriers that stop the hospitals from adding dental services.

Response: We agree with your observation. Ultimately, it is necessary to identify what policies will be effective to prepare the hospital system so that many hospitalized patients will receive appropriate dental care during their stay. Current Japanese policy encourages collaborative medical-dental teams in hospitals. However, few studies have been reported regarding the presence of dental care availability. Therefore, our study examined the presence of availability of dental care in hospitals, and related factors. Other important factors, such as public or private, for-profit or not-for-profit, regular hospitals, or research hospitals, were already evaluated in our analyses. DPC hospital types refer to the hospital function, such as academic, core, or general hospitals. All DPC type I hospitals are academic hospitals or national center hospitals. DPC type II hospitals include secondary hospitals such as affiliated hospitals of academic hospitals, and core hospitals. DPC type III hospitals are acute general hospitals. Public or profit hospitals were evaluated by hospital establisher; “the hospital founders can be divided into public (the Japanese government, public organizations, social insurance-related organizations), private (medical corporations, other corporations, others), and mixed (university) sources. Our findings, therefore, implied that public hospitals might be more likely to have working dentists and provide peri-operative oral care than private hospitals.” (as a noted manuscript. Lines 210-214).As per your thoughts, these factors were related to the implementation of medical-dental collaboration. We think the present study provides essential information for future studies to design new policies to encourage hospitals to add dental services, because this is the first study to report on the description of characteristics of the absence of dental care availability for hospitalized patients. Rather than making policy decisions based solely on experts’ opinions on how to promote medical-dental collaboration, we believe that the use of objective, secondary data to identify which hospitals are not participating in medical-dental collaboration could be a useful resource. We have added some content regarding this in the Discussion section (lines, 238-240).

Besides, the authors mentioned that " This finding helps explore methods to encourage hospitals that do not provide dental care to provide dental care in further research. ".

For now, I don't think the findings are helpful in that way. Maybe the authors can provide more background information. If a hospital is not allowed to provide dental services by certain authority/policy, or if a hospital is for-profit but adding dental services is not profitable because there are more affordable services nearby, or if a hospital is designed for certain specialty diseases that adding dental services is inherently not an interest of them, all such situations are quite likely where the manuscript's results won't help. Alternatively, the authors may filter their data to include only the hospitals without such mentioned concerns.

For the current version, I feel a significant amount of background information is still missing. The authors should work further to clarify the background information and to make sure the data and results reasonably support the conclusions.

Response: Under the Japanese medical system, all hospitals can provide dental care for in-hospital patients. Therefore, it was unnecessary to filter our data to include only the hospitals without characteristics mentioned in the reviewer’s concerns. We are not arguing that all hospitals should hire dentists to perform dental care themselves. The reasons are that, per your comment, some hospitals have more affordable services nearby, or some hospitals are designed for certain specialty diseases and adding dental services is inherently not of interest to them. As mentioned above, we argue that hospitals should provide appropriate dental care to be available to those who need it (e.g., perioperative oral care for surgery patients). If hospitals do not need to employ dentists, it may be possible to collaborate with community dental clinics to provide inpatient oral care (please see the Discussion section, lines 244-250). Unfortunately, our study initially revealed that this system has not been created in many hospitals currently not employing dentists. Thus, it is useful to consider measures to promote future medical and dental cooperation by providing the information obtained by describing the current situation.